# Association of orthostatic blood pressure response with incident heart failure: The Framingham Heart Study

Tara A. Shrout[1,2], Stephanie Pan[3], Gary F. Mitchell[4], Ramachandran S. Vasan[5,6,7,8], Vanessa Xanthakis[3,5,8]*

1 Department of Internal Medicine, Residency Program, Boston Medical Center, Boston, MA, United States of America, 2 Department of Family Medicine and Public Health, University of California San Diego, La Jolla, CA, United States of America, 3 Department of Biostatistics, Boston University School of Public Health, Boston, MA, United States of America, 4 Cardiovascular Engineering, Norwood, MA, United States of America, 5 Section of Preventive Medicine and Epidemiology, Boston University School of Medicine, Boston, MA, United States of America, 6 Department of Epidemiology, Boston University School of Public Health, Boston, MA, United States of America, 7 Boston University Center for Computing and Data Sciences, Boston, MA, United States of America, 8 Framingham Heart Study, Framingham, MA, United States of America

* vanessax@bu.edu

**Data Availability Statement:** Third-party data used in the course of this study can be accessed with a simple request from BioLINCC (https://biolincc. nhlbi.nih.gov/studies/framcohort/). The authors

## Abstract

### Importance

Orthostatic hypotension (OH) and hypertension (OHT) are aberrant blood pressure (BP) regulation conditions associated with higher cardiovascular disease risk. The relations of OH and OHT with heart failure (HF) risk in the community are unclear and there remains a paucity of data on the relations with HF subtypes [HF with reduced ejection fraction (HFrEF) and HF with preserved ejection fraction (HFpEF)].

### Objective

Relate OH and OHT with HF risk and its subtypes.

### Design

Prospective observational cohort.

### Setting

Community-based individuals in the Framingham Heart Study Original Cohort.

### Participants

1,914 participants (mean age 72 years; 1159 women) attending examination cycle 17 (1981–1984) followed until December 31, 2017 for incident HF or death.

confirm that others are able to access these data in the same manner as themselves. The authors also confirm that they had no special access privileges which others would not have.

**Funding:** SOURCES OF FUNDING The Framingham Heart Study acknowledges the support of The Framingham Heart Study (FHS) acknowledges the support of Contracts NO1-HC-25195, HHSN268201500001I and 75N92019D00031 from the National Heart, Lung and Blood Institute for this research. Dr. Vasan is supported in part by the Evans Medical Foundation and the Jay and Louis Coffman Endowment from the Department of Medicine, Boston University School of Medicine. DISCLOSURES Gary F. Mitchell is the owner of Cardiovascular Engineering, Inc., which designs and manufactures devices that measure vascular stiffness. The company uses these devices in clinical trials that evaluate the effects of diseases and interventions on vascular stiffness. He also reports receiving grants from the National Institutes of Health and Novartis and consulting fees from Novartis, Bayer, Merck, and Servier. The remaining authors declare no conflicts.

**Competing interests:** The authors have declared that no competing interests exist.

## Exposures

OH or OHT, defined as a decrease or increase, respectively, of ≥20/10 mmHg in systolic/diastolic BP upon standing from supine position.

## Outcomes and measures

At baseline, 1,241 participants had a normal BP response (749 women), 274 had OH (181 women), and 399 had OHT (229 women). Using Cox proportional hazards regression models, we related OH and OHT to risk of HF, HFrEF, and HFpEF compared to the absence of OH and OHT (reference), adjusting for age, sex, body mass index, systolic and diastolic BP, hypertension treatment, smoking, diabetes, and total cholesterol/high-density lipoprotein.

## Results

On follow-up (median 13 years) we observed 492 HF events (292 in women; 134 HFrEF, 116 HFpEF, 242 HF indeterminate EF). Compared to the referent, participants with OH [n = 84/274 (31%) HF events] had a higher HF risk (Hazards Ratio [HR] 1.47, 95% CI 1.13–1.91). Moreover, OH was associated with a higher HFrEF risk (HR 2.21, 95% CI 1.34–3.67). OHT was not associated with HF risk.

## Conclusions and relevance

Orthostatic BP response may serve as an early marker of HF risk. Findings suggest shared pathophysiology of BP regulation and HF, including HFrEF.

## Introduction

With an aging population, the number of people in the United States with heart failure (HF) is projected to exceed 8 million by 2030 [1, 2]. Early recognition of individuals at risk for HF is key to modify risk factors and improve outcomes, though it can be challenging because conventional risk factors may not always be present in those who develop HF [3, 4]. Accumulating evidence suggests that orthostatic hypotension (OH) and orthostatic hypertension (OHT), which are often asymptomatic dysregulated blood pressure (BP) responses to standing, are associated with a higher risk of cardiovascular (CVD) and all-cause mortality [5–10]. Normal compensatory responses moderate BP fluctuation upon standing; however, OH and OHT can manifest in the setting of dysregulation of heart rate, cardiac output or peripheral vascular resistance, which are important and potentially modifiable mediators in the pathogenesis of CVD, particularly HF [7, 11]. Despite the easy assessment of OH and OHT, clinical implications of these conditions on the risk of HF and HF subtypes in the community are not well understood.

Among prior cohort studies that examined the association between OH and risk of HF, most utilized endpoints of advanced disease, i.e., HF-related hospitalizations and death [12–15]. Of these reports, some observed that the association of OH with incident HF was stronger among younger participants (those <45 years of age [13] and <55 years of age [14]) while others have reported significant associations only among the older adults (>78 years of age) [16]. Furthermore, several reports note attenuation of the association between OH and HF risk after adjustment for standard risk factors including hypertension and diabetes [12–18]. Thus, a knowledge gap currently exists regarding the prognostic value of OH in relation to incident HF for individuals in the community.

Furthermore, less emphasis has been placed on the prognostic implication of OHT [9, 19] both in the clinical and academic settings, though OHT is reported to occur at similar rates as OH [20]. Recently, OHT was associated with a higher risk of end-organ damage [21] (i.e., coronary heart disease and chronic kidney disease), cerebrovascular disease [22], and mortality [23]; however, data on the relation of OHT with HF risk are scarce. Lastly, although OH is frequently concomitant with HF with reduced ejection fraction (HFrEF) [24], the risk of developing HFrEF or HF with preserved ejection fraction (HFpEF) in those with and without OH and OHT remains unknown.

We hypothesized that presence of OH and OHT are associated with a higher risk of incident HF and its subtypes. We tested this hypothesis using data from the Framingham Heart Study (FHS) Original Cohort.

## Methods

### Study sample

The FHS is a longitudinal cohort study in a community-based setting that has been previously described [25, 26]. Of the 2,144 FHS Original Cohort participants who attended examination cycle 17 (1981–1984), we excluded those with any of the following: prevalent HF at exam 17 (n = 67), missing data on orthostatic BP (n = 161), and missing data on follow-up (n = 2), resulting in a final sample size of 1,914 participants. Study protocols were approved by the Institutional Review Board at Boston University Medical Center. All participants provided written informed consent consistent with the Declaration of Helsinki. No participants received compensation or were offered incentives for participating in the present investigation.

### Assessment of orthostatic blood pressure (BP) response

Orthostatic BP responses were assessed during examination cycle 17. First, systolic and diastolic BP were obtained in the supine position after participants remained resting for five minutes. Participants were then asked to move from the supine position to a standing position. Standing BP was measured after two minutes of standing, which is considered enough time to reach physiologic equilibrium after orthostatic stress [27, 28]. OH was defined as a decrease in systolic BP (SBP) of at least 20 mm Hg or diastolic BP (DBP) of at least 10 mm Hg upon change in position from supine to standing [29]. Similarly, OHT was defined as an increase in SBP of at least 20 mm Hg or of DBP of at least 10 mm Hg upon standing from a supine position. All BP measurements were obtained by clinicians via auscultation at the level of the brachial artery using a mercury sphygmomanometer, a cuff of appropriate size and a standardized protocol.

### Outcomes of interest

The primary outcome of interest was incident HF as assessed continually after examination cycle 17 (1981–1984). HF was defined per Framingham Heart Study criteria (presence of two major, or of one major plus two minor criteria) and all HF diagnoses were reviewed and adjudicated by a panel of three experienced physicians, as previously described [30]. Briefly, the first diagnosis of HF was identified via data gathered from medical records from outpatient visits, biennial FHS follow-up visits, annual FHS telephone health history updates, and review of hospitalization and death records. The secondary outcomes of interest for this investigation were incident HF subtypes, i.e., HFrEF (HF with a left ventricular ejection fraction [EF] <50%) and HFpEF (HF with a left ventricular EF ≥50%) [31–34].

The presence of a reduced LV systolic function was determined by reviewing the hospitalization and echocardiography reports we obtain from the participants' physicians when they are hospitalized with a HF event.

## Covariates

Covariates were assessed at baseline during examination cycle 17 and included the following: age, sex, body mass index (BMI), seated SBP, seated DBP, use of antihypertensive treatment, current smoking status (ascertained using self-report and defined as smoking one or more cigarettes per day one year prior to examination cycle 17), diabetes (defined as fasting blood glucose level ≥126 mg/dL, non-fasting blood glucose level ≥200 mg/dL, or the use of glucose-lowering medication), and total cholesterol (TC)/high density lipoprotein cholesterol (HDL-C).

## Statistical analysis

We used Cox proportional hazards regression models to relate OH and OHT to risk of HF on follow-up after examination cycle 17, after confirming that the assumption of proportional hazards was met. In addition, we related OH and OHT to HF subtypes (HFrEF and HFpEF) accounting for the competing risk of the alternative subtype (HFpEF and HFrEF, respectively). We used a categorical variable as our exposure, as follows: OH, OHT, and absence of both OH and OHT (referent group). Models were adjusted for: (A) age and sex; (B) age, sex, BMI, seated SBP, seated DBP, antihypertensive treatment, smoking, diabetes, and TC/HDL-C. To investigate whether known CVD risk factors modified the relation of OH or OHT with HF risk, we evaluated the interactions of OH and OHT with age, sex, BMI, SBP, and diabetes by including the corresponding cross-product terms in separate Cox models. We created Kaplan-Meier curves to depict the relation of OH and OHT with HF risk. We also evaluated the competing risk of death in the association of OH and OHT with HF risk. For all models, a 2-sided p-value of < 0.05 was considered statistically significant. All analyses in the present investigation were conducted using SAS software version 9.4 (SAS Institute Inc, Cary, NC).

## Results

The descriptive characteristics of the study sample at examination cycle 17 are shown by sex in **Table 1** and by baseline orthostatic BP response category in **S1 Table**. Most participants were elderly, overweight, and hypertensive. We observed 274 (14%) participants with prevalent OH and 399 (21%) with prevalent OHT; the remaining participants were categorized as having a normal BP response to posture change.

## Associations of orthostatic hypotension and orthostatic hypertension with incident HF

On median follow-up of 13 years, 492 (26%) participants developed HF [292 women (59%)]. Presence of OH was associated with a 1.5-fold higher risk of HF, after adjustment for age, sex, BMI, seated SBP, seated DBP, antihypertensive treatment, smoking, diabetes, and TC/HDL-C (**Table 2**). We did not observe statistically significant interactions of OH with age, sex, BMI, SBP or diabetes. When evaluating the competing risk of death, results were similar. Kaplan-Meier plots for survival free of HF by baseline postural BP response category are shown in **Fig 1**. OHT was not significantly associated with risk of HF.

## Association of orthostatic hypotension with HFrEF and HFpEF

Among participants who developed HF, 134 developed HFrEF and 116 developed HFpEF; the remaining participants with HF did not have available data on ejection fraction. Presence of OH was associated with a 2.2-fold higher risk of developing HFrEF, adjusting for age, sex, BMI, SBP, DBP, antihypertensive treatment, smoking, diabetes, and TC/HDL-C. OH was not significantly associated with incident HFpEF.

**Table 1. Characteristics of study sample by sex.**

| | Men (n = 755) | Women (n = 1,159) |
|---|---|---|
| Age, years | 71 ± 7 | 72 ± 7 |
| Body Mass Index, kg/m$^2$ | 26.7 ± 3.7 | 26.3 ± 4.7 |
| Seated Systolic Blood Pressure, mm Hg | 141 ± 18 | 143 ± 20 |
| Seated Diastolic Blood Pressure, mm Hg | 78 ± 10 | 77 ± 10 |
| Heart rate, min$^{-1}$ | 69 ± 13 | 72 ± 13 |
| Hypertension treatment | 296 (39%) | 546 (47%) |
| TC/HDL-C, mg/dL | 5.3 ± 1.6 | 4.8 ± 1.6 |
| Lipid-lowering treatment | 6 (1%) | 22 (2%) |
| Diabetes Mellitus | 61 (8%) | 59 (5%) |
| Glucose-lowering treatment | 53 (7%) | 54 (5%) |
| Current Smoker | 125 (17%) | 218 (19%) |
| Orthostatic Hypotension (OH) | 93 (12%) | 181 (16%) |
| Orthostatic Hypertension (OHT) | 170 (23%) | 229 (20%) |
| Neither OH nor OHT | 492 (65%) | 749 (65%) |

Data are given as mean ± SD, unless otherwise indicated. TC indicates total cholesterol; HDL-C, high-density lipoprotein.

## Discussion

### Principal findings

We observed that OH was associated with a higher risk of developing HF and HFrEF, but not HFpEF. OHT was not associated with HF risk.

### Comparison with the literature

Consistent with our findings, Fedorowski et al. and Jones et al. using data from the Malmö and ARIC cohort studies, respectively, reported similar associations between OH and HF [13, 14].

**Table 2. Association of orthostatic blood pressure response with risk of HF and HF subtypes.**

| BP Response | HF (n = 492) | p-value[b] | HFrEF (n = 134) | p-value[c] | HFpEF (n = 116) | p-value[c] |
|---|---|---|---|---|---|---|
| Model | HR (95% CI) | | HR (95% CI) | | HR (95% CI) | |
| OH (n = 274) | | | | | | |
| Unadjusted | **1.78 (1.40–2.27)** | **<0.0001** | **2.21 (1.40–3.51)** | **0.0007** | 0.90 (0.49–1.66) | 0.73 |
| Age- and sex-adjusted | **1.60 (1.25–2.04)** | **0.0002** | **2.31 (1.44–3.72)** | **0.0006** | 0.85 (0.45–1.61) | 0.62 |
| Multivariable-adjusted[a] | **1.47 (1.13–1.91)** | **0.0045** | **2.21 (1.34–3.67)** | **0.002** | 0.76 (0.38–1.52) | 0.44 |
| OHT (n = 399) | | | | | | |
| Unadjusted | 1.16 (0.93–1.45) | 0.20 | 1.31 (0.87–1.98) | 0.19 | 1.01 (0.64–1.60) | 0.96 |
| Age- and sex-adjusted | 1.17 (0.94–1.46) | 0.16 | 1.29 (0.86–1.95) | 0.22 | 1.02 (0.64–1.61) | 0.95 |
| Multivariable-adjusted[a] | 1.16 (0.91–1.47) | 0.22 | 1.32 (0.86–2.03) | 0.21 | 0.88 (0.54–1.45) | 0.62 |
| Neither (n = 1,241) | Reference | --- | Reference | --- | Reference | --- |

Data are given as hazard ratio (HR) with 95% confidence intervals (CI). BP indicates blood pressure; OH, orthostatic hypotension; OHT, orthostatic hypertension; HF, heart failure; HFrEF, heart failure with reduced ejection fraction; HFpEF, heart failure with preserved ejection fraction.

[a] Multivariable-adjusted models accounted for age, sex, body mass index, seated systolic and diastolic blood pressure, hypertension treatment, smoking, diabetes, and total cholesterol/high-density lipoprotein. HFpEF is treated as a competing event in the analysis for the incident of HFrEF and vice versa.

[b] p values were calculated using the Cox-proportional hazard model.

[c] p values were calculated using Fine and Gray's subdistribution hazard model.

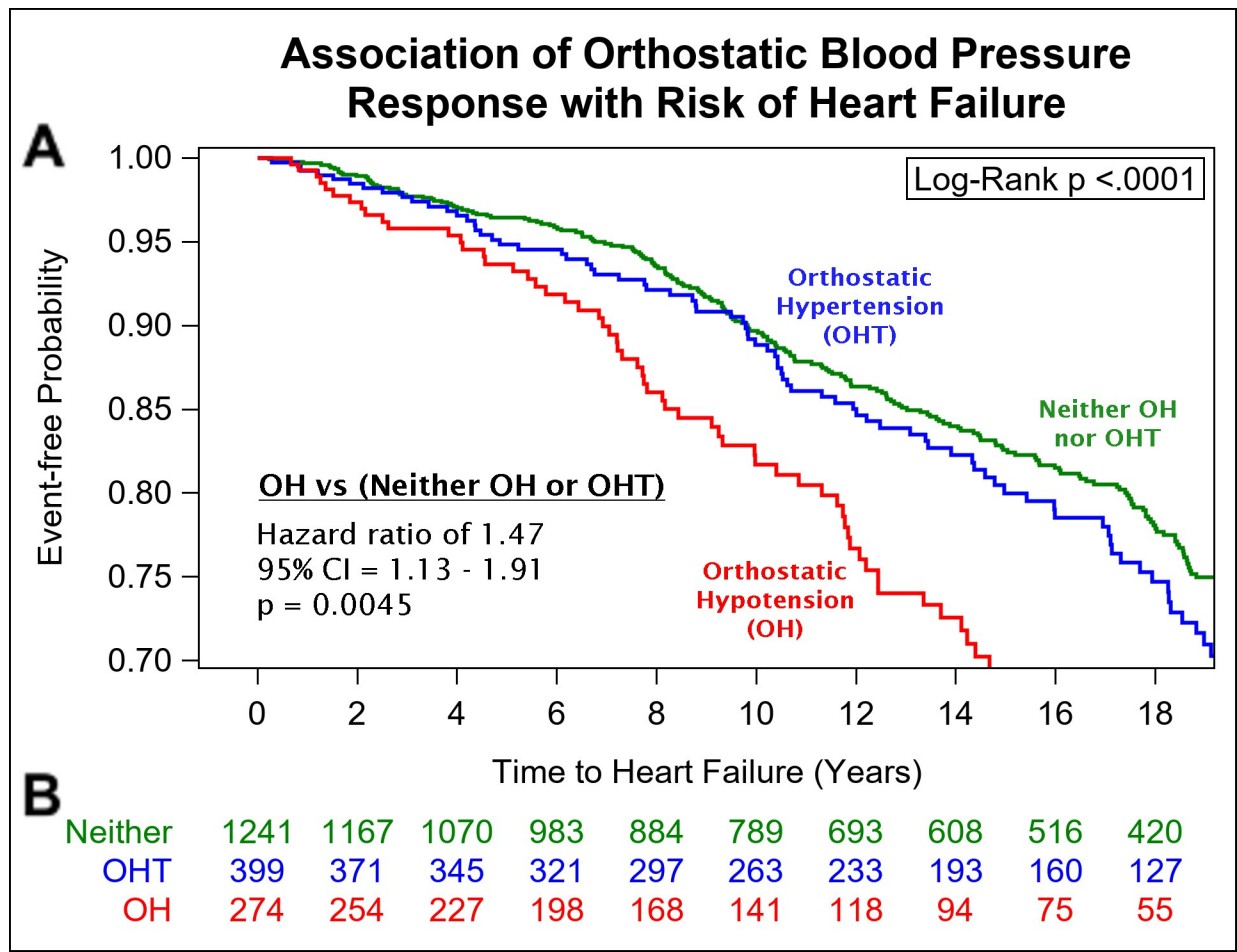

**Fig 1. Kaplan-Meier curve for heart failure risk by orthostatic blood pressure response.** (A) Presence of OH at baseline was associated with a 1.5 times higher risk of incident HF when compared to a normal (neither OH nor OHT) orthostatic BP response (multivariable-adjusted model HR 1.47, 95% CI 1.13–1.91, p = 0.0045). (B) Participants at risk per follow-up year. OH indicates orthostatic hypotension; OHT, orthostatic hypertension; HR, hazard ratio.

We observed a higher prevalence of OH and higher incidence of HF compared to the aforementioned reports, such that 26% of participants in our investigation developed HF, compared to 4% and 14% in the Malmö and ARIC studies, respectively. Our relatively elderly sample, with a mean age of 72 years, may have contributed to these findings. Additionally, compared to the Malmö and ARIC studies, which related OH to incident HF based on HF-related hospitalizations and deaths, our investigation assessed the development of HF events in the community, which may capture a wider spectrum of HF cases, including individuals with milder disease.

In contrast to prior conflicting studies in which the association of OH and risk of HF was lost or notably attenuated after adjustment for traditional risk factors [13, 14, 16, 17], our investigation observed sustained statistical significance after adjusting for all covariates (age, sex, BMI, seated SBP, seated DBP, antihypertensive treatment, smoking, diabetes, TC/HDL-C) and after adjustments for competing risk of death given our elderly cohort. Importantly, we did not observe any effect modifications of the association between OH and HF by age, sex, BMI, SBP or diabetes, thus suggesting a prognostic role of OH consistent across individuals with baseline hypertension, diabetes, elevated BMI, and among age groups in our sample. It

remains unknown if the role of OH on HF risk is modified by optimal compliance with antihypertensive therapy or medications. Data on medication compliance are not available for the present investigation; it is important to evaluate the effect of medication compliance on the observed association in future studies.

Our investigation also extends the literature with assessment for risk of HF subtypes (HFrEF and HFpEF) using echocardiographic data. OH is frequently concomitant with prevalent HFrEF [24]; we add the observation of a direct association between OH and risk of incident HFrEF, a finding that may represent shared etiologies of OH and HFrEF, including an inability to augment cardiac output in response to orthostatic stress. In contrast, OH was not significantly associated with incident HFpEF [35, 36].

Lastly, we observed that OHT was not associated with future risk of HF. Prior studies related OHT with risk of CVD, though the association with risk of HF was largely unknown. We note that we observed a much higher prevalence of OHT compared to other cohorts of similar age [20, 23], although OHT criteria varied among studies [9, 19, 22, 37]. We should note that OH could be a marker of frailty, as discussed by Liguori et al; however, we do not have frailty data available at examination cycle 17 to evaluate this hypothesis [38].

## Strengths and limitations

Strengths of the present investigation include reduction of selection bias by using a large community-dwelling sample, minimization of residual confounding with comprehensive assessment of CVD risk factors, and the long follow-up time period (maximum 35 years). Moreover, the methods to assess OH and OHT via standard orthostatic BP measurement can be easily implemented in clinic or via telemedicine visits with proper instruction.

There are several limitations that should be mentioned. First, participants in our sample were predominantly white individuals of European ancestry and elderly; as such, findings may not be generalizable to other racial/ethnic groups or to other age groups. Second, orthostatic BP measurements for OH and OHT were performed only once during examination cycle 17; consequently, participants classified as not having OH or OHT may have had BP responses within diagnostic criteria at another time point, and vice-versa [39]; such regression dilution bias may have attenuated associations in our investigation [40]. Third, the diagnostic criteria for OHT are less characterized with variable definitions used in different reports [9, 19, 22]. In fact, a recent study defined OHT as only an increase of 10 mmHg in SBP upon standing, compared to our investigation in which an increase of at least 20 mmHg in SBP or 10 mmHg in DBP was required [37]. Finally, echocardiography was not used as frequently during the period when examination cycle 17 took place, explaining the large number of participants with missing information for ejection fraction at the time of HF.

## Conclusion

Our investigation extends previous findings of associations of OH and OHT with HF risk to the community-dwelling and elderly populations, also including HFrEF and HFpEF. Routine clinical assessment of orthostatic BP responses may improve stratification for future risk of HF; specifically, OH may serve as a nontraditional factor associated with higher HF and HFrEF risk, even among those with hypertension. Further studies are warranted to investigate mechanisms underlying the observed associations.

## Supporting information

**S1 Table. Characteristics of study sample by orthostatic blood pressure response category.**
(DOCX)

## Acknowledgments

We acknowledge the dedication of the Framingham Heart Study participants without whom this research would not be possible.

## Author Contributions

**Conceptualization:** Vanessa Xanthakis.

**Formal analysis:** Stephanie Pan, Vanessa Xanthakis.

**Investigation:** Tara A. Shrout.

**Methodology:** Vanessa Xanthakis.

**Supervision:** Ramachandran S. Vasan, Vanessa Xanthakis.

**Writing – original draft:** Tara A. Shrout.

**Writing – review & editing:** Stephanie Pan, Gary F. Mitchell, Ramachandran S. Vasan, Vanessa Xanthakis.

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
