## [Decision Letter · Decision Letter 0]

7 Mar 2022

PONE-D-22-03355Association of orthostatic blood pressure response with incident heart failure: The Framingham Heart StudyPLOS ONE

Dear Dr. Xanthakis,

Thank you for submitting your manuscript to PLOS ONE. After careful consideration, we feel that it has merit but does not fully meet PLOS ONE’s publication criteria as it currently stands. Therefore, we invite you to submit a revised version of the manuscript that addresses the points raised during the review process.

We look forward to receiving your revised manuscript.

Kind regards,

Pasquale Abete

Academic Editor

PLOS ONE

Journal Requirements:

“Gary F. Mitchell is the owner of Cardiovascular Engineering, Inc., which designs and manufactures devices that measure vascular stiffness. The company uses these devices in clinical trials that evaluate the effects of diseases and interventions on vascular stiffness.  He also reports receiving grants from the National Institutes of Health and Novartis and consulting fees from Novartis, Bayer, Merck, and Servier. The remaining authors declare no conflicts.”

Additional Editor Comments:

According to Reviewers' decision, the manuscript need a major revision.

Reviewers' comments:

Reviewer's Responses to Questions

**Comments to the Author**

1. Is the manuscript technically sound, and do the data support the conclusions?

Reviewer #1: Yes

Reviewer #2: Yes

2. Has the statistical analysis been performed appropriately and rigorously? 

Reviewer #1: Yes

Reviewer #2: Yes

3. Have the authors made all data underlying the findings in their manuscript fully available?

Reviewer #1: Yes

Reviewer #2: Yes

4. Is the manuscript presented in an intelligible fashion and written in standard English?

Reviewer #1: Yes

Reviewer #2: Yes

5. Review Comments to the Author

Reviewer #1: The study evaluates the effect of orthostatic hypotension and hypertension with heart failure risk in the community and the relations with HF subtypes [HF with reduced ejection fraction and HF with preserved ejection fraction]. Data derived from 1,914 participants enrolled in the 17 cycle between 1981–1984 of the Framingham Heart Study Original Cohort followed until December 31, 2017 for incident HF or death. OH or OHT, were defined as a decrease or increase, respectively, of 20/10 mmHg in systolic/diastolic BP upon standing from supine position.At baseline, 1,241 participants had a normal BP response, 274 had OH, and 399 had OHT. On follow-up (median 13 years) we observed 492 HF events (292 in women;134 HFrEF, 116 HFpEF, 242 HF indeterminate EF). Compared to the referent,

participants with OH [n= 84/274 (31%) HF events] had a higher HF risk (Hazards Ratio [HR] 1.47, 95% CI 1.13—1.91). Moreover, OH was associated with a higher HFrEF risk (HR 2.21, 95% CI 1.34—3.67). OHT was not associated with HF risk.

The topic is important and the manuscript is relevant even if is confirmative of a known association. The study methodology is eccelent. Of interest is the asoociation of OH with HFrEF. Data supports conclusions, the manuscript is well written and I have no suggestions to improve it.

Reviewer #2: Orthostatic hypotension (OH) and hypertension (OHT) are aberrant blood pressure regulation conditions, expression of dysregulation of heart rate, cardiac output or peripheral vascular resistance, that are associated with higher cardiovascular disease risk. Authors evaluated the role of OH and OHT in HF risk founding that OH was associated with a higher risk of developing HF and HFrEF, but not HFpEF. In contrast, OHT was not associated with HF risk.

The paper has merit, but there are points that deserve a major revision.

1. In “Outcomes of Interest” section, no information is given about definition of incident HF subtypes. Please describe how the presence of a reduced LV systolic function was determined by echocardiography.

2. The data on ejection fractions of 242 patients with HF are missing. Considering that these represent 49% of the population with HF emphasize this important limitation of the study.

3. As reported by the authors, the association between OH and HF has already been described, but in contrast to prior studies in which the association of OH and risk of HF was lost or attenuated after adjustment for traditional risk factors, investigators observed in this work higher incidence of HF and sustained statistical significance after adjusting for all covariates (age, sex, BMI, seated SBP, seated DBP, antihypertensive treatment, smoking, diabetes, etc). Please describe the clinical and demographic differences in the sample in relation to the BP response to posture change (OH, OHT, neither OH nor OHT)

4. As reported, the elderly sample, with a mean age of 72 years, may have contributed to higher prevalence of OH and higher incidence of HF. What about frailty status? Please see and discuss “Liguori I. et al. Orthostatic Hypotension in the Elderly: A Marker of Clinical Frailty? J Am Med Dir Assoc. 2018 Sep;19(9):779-78”.

5. Authors suggest that OH may serve as a non traditional factor associated with higher HF and HFrEF risk, even among patient with hypertension. Please discuss if the effect of OH on HF could depend on a non-optimal use of anti-hypertensive therapy.

6. PLOS authors have the option to publish the peer review history of their article (what does this mean?). If published, this will include your full peer review and any attached files.

Reviewer #1: **Yes: **Francesco Cacciatore

Reviewer #2: No

---

## [Author Response · Author response to Decision Letter 0]

18 Mar 2022

See Response to Reviewers document.

---

## [Decision Letter · Decision Letter 1]

1 Apr 2022

Association of orthostatic blood pressure response with incident heart failure: The Framingham Heart Study

PONE-D-22-03355R1

Dear Dr. XANTHAKIS,

We’re pleased to inform you that your manuscript has been judged scientifically suitable for publication and will be formally accepted for publication once it meets all outstanding technical requirements.

Kind regards,

Pasquale Abete

Academic Editor

PLOS ONE

Additional Editor Comments (optional):

No further comments

Reviewers' comments:

Reviewer's Responses to Questions

**Comments to the Author**

1. If the authors have adequately addressed your comments raised in a previous round of review and you feel that this manuscript is now acceptable for publication, you may indicate that here to bypass the “Comments to the Author” section, enter your conflict of interest statement in the “Confidential to Editor” section, and submit your "Accept" recommendation.

Reviewer #1: All comments have been addressed

Reviewer #2: All comments have been addressed

2. Is the manuscript technically sound, and do the data support the conclusions?

Reviewer #1: Yes

Reviewer #2: Yes

3. Has the statistical analysis been performed appropriately and rigorously? 

Reviewer #1: Yes

Reviewer #2: Yes

4. Have the authors made all data underlying the findings in their manuscript fully available?

Reviewer #1: Yes

Reviewer #2: Yes

5. Is the manuscript presented in an intelligible fashion and written in standard English?

Reviewer #1: Yes

Reviewer #2: Yes

6. Review Comments to the Author

Reviewer #1: All comments have been addressed and in my first revision I found the manuscript of interest. The present revision have improved the messagge of the paper.

Reviewer #2: The manuscript is really improved. All questions arised have been answered. the manusript is now acceptable to be published in PONE.

7. PLOS authors have the option to publish the peer review history of their article (what does this mean?). If published, this will include your full peer review and any attached files.

Reviewer #1: No

Reviewer #2: No

---

## [Editor Report · Acceptance letter]

13 Apr 2022

PONE-D-22-03355R1 

Association of orthostatic blood pressure response with incident heart failure: The Framingham Heart Study 

Dear Dr. Xanthakis:

I'm pleased to inform you that your manuscript has been deemed suitable for publication in PLOS ONE. Congratulations! Your manuscript is now with our production department. 

Kind regards, 

on behalf of

Prof. Pasquale Abete 

Academic Editor

PLOS ONE